# Predicting Future Actions of Reinforcement Learning Agents

**Stephen Chung**[*]
University of Cambridge

**Scott Niekum**
University of Massachusetts Amherst

**David Krueger**
Mila

## Abstract

As reinforcement learning agents become increasingly deployed in real-world scenarios, predicting future agent actions and events during deployment is important for facilitating better human-agent interaction and preventing catastrophic outcomes. This paper experimentally evaluates and compares the effectiveness of future action and event prediction for three types of RL agents: explicitly planning, implicitly planning, and non-planning. We employ two approaches: the inner state approach, which involves predicting based on the inner computations of the agents (e.g., plans or neuron activations), and a simulation-based approach, which involves unrolling the agent in a learned world model. Our results show that the plans of explicitly planning agents are significantly more informative for prediction than the neuron activations of the other types. Furthermore, using internal plans proves more robust to model quality compared to simulation-based approaches when predicting actions, while the results for event prediction are more mixed. These findings highlight the benefits of leveraging inner states and simulations to predict future agent actions and events, thereby improving interaction and safety in real-world deployments.

## 1 Introduction

As reinforcement learning (RL) becomes increasingly applied in the real world, ensuring the safety and reliability of RL agents is paramount. Recent advancements have shown that agents can exhibit complex behaviors, making it crucial to understand and anticipate their actions. This is especially important in scenarios where misaligned objectives [1] or unintended consequences could result in suboptimal or even harmful outcomes. For instance, consider an autonomous vehicle controlled by an RL agent that might unpredictably decide to run a red light to optimize travel time. Predicting this behavior in advance would enable timely intervention to prevent a potentially dangerous situation. This capability is also beneficial in scenarios that require effective collaboration and information exchange among multiple agents [2–4]. For example, if passengers and other drivers know whether a self-driving car will turn left or right, it becomes much easier and safer to navigate the roads. Thus, the ability to accurately predict an agent's future behavior can help reduce risks and ensure smooth interaction between agents and humans in real-world situations.

In this paper, we explore the task of predicting future actions and events when deploying a trained agent, such as whether an agent will turn left in five seconds. The distribution of future actions and events cannot be computed directly, even with access to the policy, because the future states are unknown. We consider two methods for predicting future actions and events: the inner state approach and the simulation-based approach. We apply these approaches to agents trained with various RL algorithms to assess their predictability[1].

---

[*]Correspondence to: `mhc48@cam.ac.uk`

[1]Full code is available at `https://github.com/stephen-chung-mh/predict_action`

38th Conference on Neural Information Processing Systems (NeurIPS 2024).

In the *inner state approach*, we assume that we have full access to the *inner state* of the agent during deployment. Here, the inner state refers to all the intermediate computations required to determine the final action executed by the agent, such as the simulation of the world model for explicit planning agents or the hidden layers for agents parametrized by deep neural networks. We seek to answer the following questions: (i) How informative are these inner states for predicting future actions and events? (ii) How does the predictability of future actions and events vary across different types of RL agents with different inner states?

As an alternative to the inner state approach, we explore a *simulation-based approach* by unrolling the agent in a learned world model and observing its behavior. Assuming we have a sufficiently accurate world model that resembles the real environment, this simulation should provide valuable information for predicting future actions and events in the real environment. We seek to answer the following question: (iii) How do the performance and robustness of the simulation-based approach compare to the inner state approach in predicting future actions and events across different agent types?

We conduct extensive experiments to address the above research questions. To summarize, the main contributions of this paper include:

1. To the best of our knowledge, this is the first work to formally compare and evaluate the predictability of different types of RL agents in terms of action and event prediction.

2. We propose two approaches to address this problem: the inner state approach and the simulation-based approach.

3. We conduct extensive experiments to evaluate the effectiveness and robustness of these approaches across different types of RL agents, demonstrating that the plans of explicitly planning agents are more informative for prediction than other types of inner states.

## 2   Background and Notation

We consider a Markov Decision Process (MDP) defined by a tuple $(\mathcal{S}, \mathcal{A}, P, R, \gamma, d_0)$, where $\mathcal{S}$ is a set of states, $\mathcal{A}$ is a finite set of actions, $P : \mathcal{S} \times \mathcal{A} \times \mathcal{S} \rightarrow [0, 1]$ is a transition function representing the dynamics of the environment, $R : \mathcal{S} \times \mathcal{A} \rightarrow \mathbb{R}$ is a reward function, $\gamma \in [0, 1]$ is a discount factor, and $d_0 : \mathcal{S} \rightarrow [0, 1]$ is an initial state distribution. Denoting the state, action, and reward at time $t$ by $S_t$, $A_t$, and $R_t$ respectively, $P(s, a, s') = \Pr(S_{t+1} = s'|S_t = s, A_t = a)$, $R(s, a) = \mathbb{E}[R_t|S_t = s, A_t = a]$, and $d_0(s) = \Pr(S_0 = s)$, where $P$ and $d_0$ are valid probability mass functions. An episode is a sequence of $(S_t, A_t, R_t)$, starting from $t = 0$ and continuing until reaching the terminal state, a special state where the environment ends. Letting $G_t = \sum_{k=t}^{\infty} \gamma^{k-t} R_k$ denote the infinite-horizon discounted return accrued after acting at time $t$, an RL algorithm attempts to find, or approximate, a *policy* $\pi : \mathcal{S} \times \mathcal{A} \rightarrow [0, 1]$, such that for any time $t \geq 0$, selecting actions according to $\pi(s, a) = \Pr(A_t = a|S_t = s)$ maximizes the expected return $\mathbb{E}[G_t|\pi]$.

In this paper, *planning* refers to the process of interacting with an environment simulator or a world model to inform the selection of subsequent actions. Here, a world model is a learned and approximated version of the environment. We classify an agent, which is defined by its policy, into one of the following three categories based on the RL algorithm by which it is trained:

**Explicit Planning Agents.** In explicit planning agents, an environment simulator or a world model is used explicitly for planning. We consider two explicit planning agents in this paper, MuZero [5] and Thinker [6], given their superior ability in planning domains. MuZero is a state-of-the-art model-based RL algorithm that combines a learned model with Monte Carlo Tree Search (MCTS) [7, 8] for planning. During planning, MuZero uses the learned model to simulate future trajectories and performs MCTS to select the best action based on the predicted rewards and values. Thinker is a recently proposed approach that enables RL agents to autonomously interact with and use a learned world model to perform planning. The key idea of Thinker is to augment the environment with a world model and introduce new actions designed for interacting with the world model. MuZero represents a handcrafted planning approach, while Thinker represents a learned planning approach.

**Implicit Planning Agents.** In implicit planning agents, there is no learned world model nor an explicit planning algorithm, yet these agents still exhibit planning-like behavior. A notable example is the Deep Repeated ConvLSTM (DRC) [9], which excels in planning domains. DRC agents are trained

by actor-critic algorithms [10] and employ a unique architecture based on convolutional-LSTM with internal recurrent steps. The authors observe that the trained agents display planning-like properties, such as improved performance with increased computational allowance, and so argue that the agent learns to perform model-free planning.

**Non-planning Agents.** In non-planning agents, there is neither a learned world model nor an explicit planning algorithm, and these agents do not exhibit planning-like behavior. Typically, these agents perform poorly in planning domains. Examples include most model-free RL algorithms, such as the actor-critic and Q-learning. In this paper, we focus exclusively on IMPALA [10], a variant of the actor-critic algorithm, chosen for its computational efficiency and popularity.

We believe that this distinction between RL agents, adopted from previous work [11], is useful for investigating their predictability. We hypothesize that the plan made by an explicit planning agent should be more informative of future actions or events than that of an implicit planning agent, as the plan in an explicit planning agent is typically human-interpretable, whereas the plan for an implicit planning agent is stored in hidden activations. Nevertheless, the computation in these two types of agents provides indications of their future actions and thus should carry more information than the hidden activations of a non-planning agent, which lacks future plans and may merely serve as a more compact representation of the state.

## 3 Problem Statement

Given a fixed policy $\pi$, we aim to estimate the distribution of a function of the future trajectory. For example, we may want to estimate the probability of an agent entering a particular state or performing a specific action within a certain horizon. Mathematically, let $H_t = (S_t, A_t, R_t)$ denote the transition at step $t$, and let $H_{t:T} = \{H_t, H_{t+1}, \ldots, H_T\}$ denote the future trajectory from step $t$ to the last step $T$. Let $\mathcal{H}$ denote the set of all possible future trajectories. We are interested in estimating the distribution of a random variable $f(H_{t:T})$ conditioned on the current state and action:

$$\mathbb{P}(f(H_{t:T}) \mid S_t, A_t), \tag{1}$$

where $f : \mathcal{H} \rightarrow \mathbb{R}^m$ is a function specifying the variables to be predicted.

This paper focuses on predicting two particular types of information. The first type is *future action prediction*, where we want to predict the action of the agent in $L$ steps, i.e., $f(H_{t:T}) = (A_{t+1}, A_{t+2}, \ldots, A_{t+L})$ and the problem becomes estimating:

$$\mathbb{P}(A_{t+1}, A_{t+2}, \ldots, A_{t+L} \mid S_t, A_t). \tag{2}$$

An example of action prediction is whether an autonomous vehicle is going to turn left or right in the next minute. The second type is *future event prediction*, where we want to estimate the probability of a binary indicator $g : (\mathcal{S}, \mathcal{A}) \rightarrow \{0, 1\}$ being active within $L$ steps, and the problem becomes estimating:

$$\mathbb{P}\left( \bigcup_{k=1}^{L} g(S_{t+k}, A_{t+k}) = 1 \,\middle|\, S_t, A_t \right), \tag{3}$$

which is equivalent to the case $f(H_{t:T}) = \max\{ g(S_{t+k}, A_{t+k}) \}_{k=1,\ldots,L}$. In other words, (3) is the probability of the event defined by $g$ occurring within $L$ steps. An example of event prediction is predicting whether an autonomous vehicle will run a red light within a minute.

Event prediction shares resemblance to the generalized value function [12], where $f(H_{t:T}) = \sum_{k=0}^{\infty} \gamma^k g(S_{t+k}, A_{t+k})$ and we estimate its expectation $\mathbb{E}[f(H_{t:T}) \mid S_t, A_t]$. When $g$ is a binary indicator, this expectation is equivalent to the discounted sum of the probabilities of the event defined by $g$. This is arguably harder to interpret than (3); for example, it can be larger than 1 and thus is not a valid probability.

To learn these distributions, we assume access to some transitions generated by the policy $\pi$ as training data. The transitions may come from multiple episodes. In the case of future event prediction, we assume that $g(S_t, A_t)$ is also known for each transition. We further assume that the policy $\pi$ and the inner computation for each action $A_t$ within $\pi$ is known.

In this work, we assume that $\pi$ is already a trained policy and is fixed. This is the case where the agent is already deployed, and transitions during the deployment are collected. In cases where the training and deployment environments are similar, we can also use the transitions when training the agent as training data for predicting future actions and events, but this is left for future work.

# 4   Methods

Since we already have the state-action $(S_t, A_t)$ and the target output $f(H_{t:T})$ in the training data, we can treat the problem as a supervised learning task.[2] In particular, We can train a neural network that takes the state-action pair as input to predict $f(H_{t:T})$. This network is trained using gradient descent on cross-entropy loss.

Besides the state-action, there can be additional information that may help the prediction. For example, the inner computation of the policy $\pi$ may contain plans that are informative of the agent's future actions, especially in the case of an explicit planning agent. We refer to this information that is available before observing the next state $S_{t+1}$ as auxiliary information and denote it as $I_t$. We will consider two types of auxiliary information: inner states and simulations.

## 4.1   Inner State Approach

In the inner state approach, we consider choosing the agent's inner state as the auxiliary information. Here, the inner state refers to all the intermediate computations required to compute the action $A_t$. As inner states are different across different types of agents and may not all be useful, we consider the following inner state to be included in the auxiliary information:

1. MuZero: Since MuZero uses MCTS to search in a world model and selects action with the largest visit count, we select the most visited rollout as the auxiliary information. Rollouts here refer to the simulation of the world model and are composed of a sequence of transitions $(\hat{S}_{t+l}, \hat{A}_{t+l}, \hat{R}_{t+l})_{1 \leq l \leq L}$. It should be noted that the agent may not necessarily select the action sequence of this rollout, as the MCTS is performed at every step, and the search result at the next step may yield different actions. We do not use all rollouts, as MCTS usually requires many rollouts.

2. Thinker: We select all rollouts and tree representations during planning as the auxiliary information. We do not choose a particular rollout because, unlike MCTS, in Thinker, it is generally unknown which action the agent will select at the end, and Thinker usually requires only a few rollouts.

3. DRC: We select the hidden states of the convolutional-LSTM at every internal tick as the inner state, as it was proposed that the hidden state contains plans to guide future actions [9].

4. IMPALA: We select the final layer of the convolutional network as the inner state, as it is neither too primitive which may only be processing the state, nor too refined which may only contain information for computing the current action and values.

Experiment results on the alternative choices of inner states can be found in Appendix D.

## 4.2   Simulation-based Approach

As an alternative to the inner state approach, we can train a world model concurrently with the agent. Once trained, we can simulate the agent in this world model using the trained policy $\pi$ to generate rollouts. These rollouts can then be utilized as auxiliary information for the predictor. In this paper, we consider using the world model proposed in Thinker [6], which is an RNN that takes the current state and action sequence as inputs and predicts future states, rewards, and other relevant information. For both implicit and non-planning agents, the world model is trained in parallel with the agents but is not used during their selection of actions. Instead, the world model is solely employed to generate rollouts as auxiliary information for the predictors.

If the learned world model closely resembles the real environment, we expect these rollouts to yield valuable information for predicting future actions and events, as the agent's behavior in the world model should be similar to its behavior in the actual environment. In the ideal case where the world model perfectly matches the true environment dynamics, we could compute the exact future action and event distribution without needing any prediction. However, we do not consider this scenario in the paper, as this assumption is impractical for most settings.

---

[2]Temporal-difference methods are not directly applicable here, as both the action and event prediction tasks involve a limited horizon $L$ and do not sum over variables.

It should be noted that in the simulation-based approach, the world model must always predict the same state space as the input to the agent, enabling the simulation of the agent within the world model. Since agents typically receive raw state inputs (with exceptions such as Dreamer [13], where agents receive abstract state inputs), the world model should also make predictions in the raw state space rather than in a latent state space.. Consequently, the simulation-based approach may not be suitable for situations where learning a world model in the raw state space is challenging, such as predicting camera input in real-world autonomous driving scenarios.

## 5    Experiments

We conduct three sets of experiments to evaluate the effectiveness and robustness of the discussed approaches. First, we apply the inner state approach to predict future actions and events. We compare it to the case where only state-action information is provided to the predictor so as to evaluate the benefits of the proposed inner state in the prediction. Second, we apply the simulation-based approach and compare it with the inner state approach to evaluate the benefits of these two different types of auxiliary information. Finally, we consider a model ablation setting, where we deliberately make the world model inaccurate to see how the different approaches perform under such conditions.

We consider the Sokoban environment, where the goal is to push all four boxes into the four red-bordered target spaces as illustrated in Fig 1. We choose this environment because (i) a wide range of levels in Sokoban make action and event prediction challenging, and we can evaluate the predictors on unseen levels to evaluate their generalization capability; (ii) there are multiple ways of solving a level; (iii) Sokoban is a planning-based domain, so it may be closer to situations where we want to discern plans of agents in more complex settings.

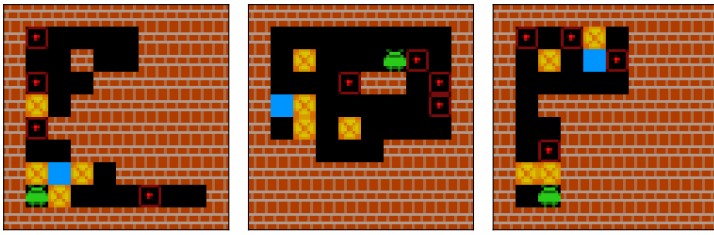

Figure 1: Example levels of Sokoban, where the goal is to push all four boxes into the four red-bordered target spaces. A box can only be pushed, not pulled, making the level irrecoverable if the boxes get stuck. We paint a random empty space blue (which still acts as an empty tile) and predict whether the agent will stand on the blue location within 5 steps.

We choose the prediction horizon $L$ to be 5 in all experiments. For action prediction, we try to predict the next five actions $A_{t+1}, A_{t+2}, ..., A_{t+5}$. For event prediction, we randomly select an empty tile in the level and paint it blue. That blue tile acts as an empty tile to the agent and serves no special function. We define the event $g$ that is to be predicted as the case where the agent stands on the blue location. In other words, we try to predict whether the agent will go to that blue location within $L$ steps.

We train four different agents using MuZero, Thinker, DRC, and IMPALA. All agents are trained for 25 million transitions. To ensure that the result would not be affected by the particular choice of the world model, we uniformly employ the world model architecture proposed in Thinker, as the world model in Thinker predicts the raw state and is suitable for both MuZero and the simulation-based approach. We train a separate world model for each agent. For DRC and IMPALA, the world model is not needed for the policy and will only be used in the predictors in the simulation-based approach.

After training the agents, we generate 50k transitions, where part or all of it will be used as the training data for the predictors. We evaluate the performance of predictors with varying training data sizes: 1k, 2k, 5k, 10k, 20k, 50k. We also generate 10k transitions as a testing dataset. For simplicity, we use greedy policies, where we select the action with the largest probability instead of sampling. The predictor uses a convolutional network to process all image information, including the current state and states in rollouts (if they exist). The encoded current state, along with other auxiliary information such as encoded states, rewards, and actions in rollouts (if they exist), will be passed to a

three-layer Transformer encoder [14], and the final layer predicts the next $L$ actions or the probability of the event within $L$ steps. More details about the experiments can be found in Appendix A.

## 5.1 Inner State Approach

Figure 2 presents the final accuracy of action prediction and the F1 score of event prediction using the inner state approach. The error bars represent the standard deviation across three independently trained predictors. The accuracy here refers to the percentage of correctly predicting all the next five actions, with no credits awarded if any action is predicted incorrectly. The graph also shows the performance of the predictors when they only receive the current state $S_t$ and action $A_t$ as inputs, as indicated by 'baseline'. Several observations can be made.

First, when access to the plans is available, the prediction accuracy for both action and event is significantly higher for explicit planning agents. For example, with 50k training data, the action prediction accuracy of the MuZero agent increases from 40% to 87% when given access to the plans. Agents using handcrafted planning algorithms (MuZero) or learned planning algorithms (Thinker) show similar performance gains. This is perhaps not surprising, as these explicit planning agents tend to follow the plans either by construction or by learning, and the explicit nature of planning facilitates easy interpretation of the plans.

Second, the case for implicit planning agents (DRC) and non-planning agents (IMPALA) is more nuanced. For action prediction accuracy, both receive a moderate improvement from accessing the hidden state. There are two possible explanations: (i) plans of the agents are stored in the learned representations that are informative of future actions; (ii) the hidden states or hidden layers contain a latent representation that is easier to learn from, compared to the raw states. To discern between the two cases, interpreting the nature and the underlying circuit of the inner states is required. We leave this to future work as interpretability is outside the scope of this paper.

Third, in contrast to action prediction, the inner state does not improve event prediction for DRC and IMPALA, likely because the blue location in the environment does not affect the reward, and the agent may ignore it in its representation. This suggests an advantage of explicit planning agents, as in explicit planning agents, we explicitly train the world model and can train it to attend not just to the reward-relevant features but to all features (or features we deem useful) in the environment. This may be important for cases where the reward function is not well designed, leading to the agent ignoring certain features that are, in fact, important to us.

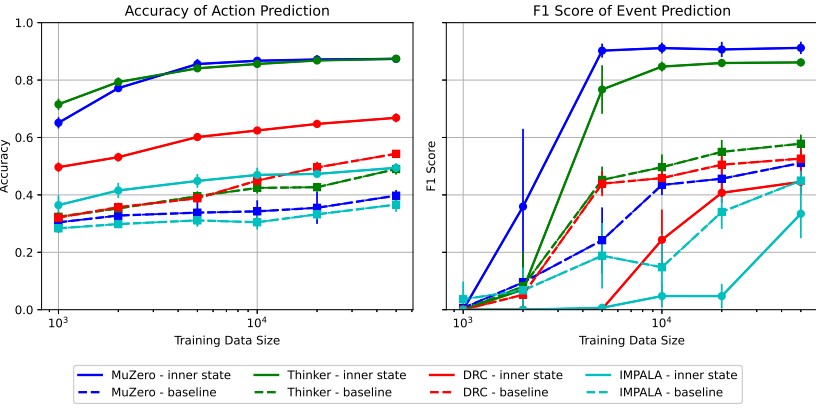

Figure 2: Final accuracy of action prediction and F1 score of event prediction with inner state approach on the testing dataset. The error bar represents two standard errors across 9 seeds.

## 5.2 Simulation-based Approach

We now consider applying the simulation-based approach to both implicit planning (DRC) and non-planning agents (IMPALA). We unroll the world model for $L = 5$ steps using the current policy and input this rollout as auxiliary information to the predictors. We can use a single rollout, as both the policy and the chosen world model are deterministic, so all rollouts will be the same.

We do not apply the simulation-based approach to explicit planning agents because (i) rollouts already exist as an inner state within the agent and can be input to the predictor, and (ii) it requires training a world model that can be unrolled for $2L$ steps instead of only $L$ steps, as the agent needs to perform planning on every step in the simulated rollout. For a fair comparison, we assume we can only train a world model that is unrolled for $L$ steps in all setups.

Figure 3 shows the final accuracy of action prediction and the F1 score of event prediction for the simulation-based approach of DRC and IMPALA. For easy comparison, we also include the result of the inner state approach of explicit planning agents in the figure.

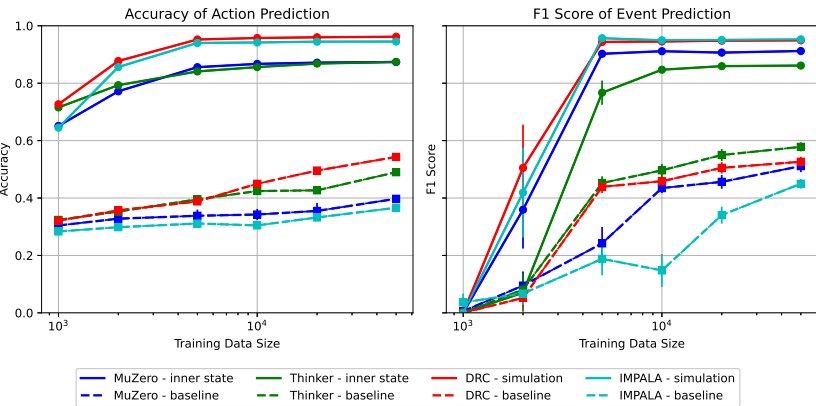

Figure 3: Final accuracy of action prediction and F1 score of event prediction with simulation-based approach (DRC and IMPALA) on the testing dataset. The absolute performance can be found in Appendix C. The error bar represents two standard errors across 9 seeds.

We observe that the predictors for DRC and IMPALA agents in the simulation-based approach perform very well, with performance surpassing that of the explicit planning agents with the inner state approach. This is because the world model we trained is very close to the true environment, so the behavior in the rollout is almost equivalent to that in the real environment. The high-quality world model also enables accurate prediction of when the agent will stand on the blue location, resulting in excellent event prediction performance.

## 5.3 World Model Ablation

Learning an accurate world model may not be feasible in some settings, such as auto-driving in the real world. An inaccurate world model will affect the plan quality of explicit planning agents, rendering the plan less informative in the inner state approach. An inaccurate world model will also affect the quality of rollouts in the simulation-based approach, leading to inconsistent behaviour between rollouts and real environments. As such, it is important to understand how the inner state approach and simulation-based approach differ when the learned world model is not accurate.

To investigate this, we designed three different settings where learning an accurate world model is challenging. In the first setting, we use a world model with a much smaller size, making it more prone to errors. In the second setting, we randomly replace the agent's action with a no-operation 25% of the time, introducing stochastic dynamics into the environment. However, since the world model we use is deterministic, it cannot account for such stochastic transitions and will yield errors. In the third setting, we consider a partially-observable Markov decision process (POMDP) case, where we randomly display the character at a position within one step of the true character location. As the world model we use only observes the current state, this will lead to uncertainty over both the true character location and the displayed character location. We repeat the above experiments in these three different settings.

Figure 4 shows the change in the final accuracy of action prediction and the F1 score of event prediction for the model ablation settings compared to the default setting. We observe that in terms of action prediction, the accuracy generally drops less in the inner state approach of explicit planning agents than in the simulation-based approach of the two other agents. This is likely because planning

does not necessitate an accurate world model, as plans can still be made without the ability to perfectly predict the future. For example, in MCTS, only values and rewards need to be predicted well, but not the state. In contrast, if we simulate an agent in a poor world model, the agent may be confused as the states may be out of distribution and never encountered during training. This leads to inconsistent behavior compared to the agent's behavior in the real environment.

In contrast, the results for event prediction are more nuanced, with the inner state approach sometimes performing better and the simulation-based approach performing better at other times. We conjecture that because the world model is not accurate, the event under consideration is often not predicted correctly. As such, more informative plans that can predict future actions do not help in event prediction, leading to mixed results.

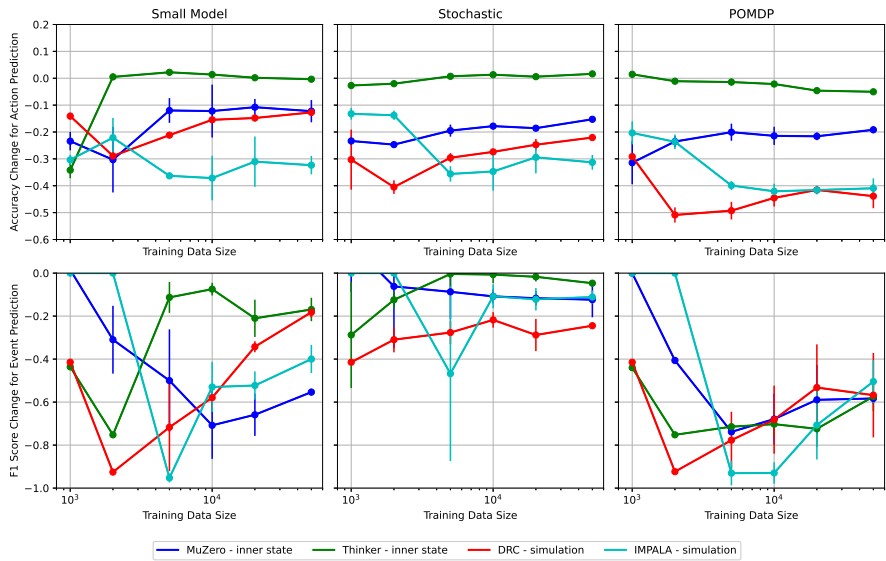

Figure 4: Change in the final accuracy of action prediction and F1 score of event prediction for the world model ablation settings. The error bar represents two standard errors across 3 seeds.

# 6   Related Works

**Safe RL:** Our work is related to safe RL, where we try to train an agent to maximize rewards while satisfying safety constraints during the learning and/or deployment processes [15]. A wide variety of methods have been proposed in safe RL, such as shielding, where one manually prevents actions that violate certain constraints from being executed [16], and Constrained Policy Optimization (CPO), which performs policy updates while enforcing constraints throughout training [17]. Many works in safe RL are based on correcting the action when deemed unsafe. Dalal et al. [18] fit a linear model to predict the violation of constraint functions and use it to correct the policy; Cheng et al. [19] project the learned policy to safe policy based on the barrier function; Thananjeyan et al. [20] guide the agent back to learned recovery zones when it is predicted that the state-action pair will lead to unsafe regions; Thomas et al. [21] use world models to predict unsafe trajectories and change the rewards to penalize safety violations.

In contrast to the works in safe RL, we are solely interested in predicting future actions and events of trained agents. The actions or events do not necessarily need to be unsafe. In the case of unsafe action or event prediction, our work allows for preemptive interruption of the deployed agent, which can be used as a last resort in addition to the above safety RL works.

**Opponent Modelling in Multi-agent Setting:** In a multi-agent setting, modeling the opponent's behavior may be beneficial in both competitive and cooperative scenarios. He et al. [22] use the opponent's inner state to better predict Q-values in a multi-agent setting. Foerster et al. [23] update an agent's policy while accounting for its effects on other agents, and Raileanu et al. [24] predict the

other agent's actions based on the same network that outputs the agent's own action. In contrast to these works, our research involves predicting agent actions multiple steps ahead and does not involve a multi-agent setting or learning a policy.

**Predictability for Human-Agent Interaction:** Recent research has highlighted the importance of predictability in enhancing human-agent interaction and collaboration. The agents in these studies are not necessarily RL agents but are often hardcoded to follow certain rules. Daronnat et al. [3] demonstrated that higher predictability in agent behavior facilitates better human-agent interaction and collaboration, particularly in real-time scenarios. Dragan et al. [2] found that legible motion, which makes an agent's intent clear, leads to more fluent human-robot collaboration. Kandul et al. [25] found that humans are better at predicting human performance than agent performance, raising concerns about human control over agents in high-stakes environments. Finally, Ahrndt et al. [4] discussed the significance of mutual predictability in human-agent teamwork. These works support the motivation that predictability of agents is an important concern for human-agent interaction.

# 7    Conclusion

In this paper, we investigated the predictability of future actions and events for different types of RL agents. We proposed and evaluated two approaches for prediction: the inner state approach and the simulation-based approach. The simulation-based approach performs well with an accurate world model but is less robust when the world model quality is compromised. Conversely, the performance of the inner state approach depends on the type of inner states and the agents. Internal plans of explicit planning agents are particularly useful compared to other types of inner states. These findings highlight the importance of leveraging auxiliary information to predict future actions and events. Enhanced predictability could lead to more reliable and safer deployment of RL agents in critical real-world applications such as autonomous driving, robotics, and healthcare, where understanding and anticipating agent behavior is important for safety and effective human-agent interaction.

Future research directions include extending our analysis to more diverse environments and RL algorithms, exploring safety mechanisms to modify agent behavior based on action prediction, and developing RL algorithms that are both predictable and high-performing.

### Limitation

The paper only evaluates the proposed approaches in a limited set of environments. Including additional environments would provide a better understanding of agent predictability, but this requires finding or designing new benchmark environments with diverse states. Additionally, the paper focuses on only four different RL algorithms. Evaluating a broader range of RL algorithms could allow for better comparisons of their predictability.

### Broader Impact Statement

This work involves predicting the actions and events of trained agents during deployment. It is important to consider the risk of false alarms, where the predictor forecasts that an agent is going to perform an unsafe action, but in fact, the agent would not be doing it. This may lead to improper responses (such as shutting down the agent) that are not warranted.

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

# A  Agent Details

In this section, we describe the details of how we trained the four types of agents discussed in the paper. Most of them follow the procedure outlined in the original paper:

1. MuZero: We use the same agent configuration as in the original paper, except for the world model, where we adopt the architecture and training method proposed in Thinker. This is to ensure that the results are not affected by the choice of the world model. We conducted 100 simulations for each search[3].

2. Thinker: We use the same default agent as described in the original paper.

3. DRC: We use the DRC(3,3) described in the original paper.

4. IMPALA: We use the large architecture but omit the LSTM component described in the original paper.

All the hyperparameters are consistent with those in the original paper, and the agents are all trained using 25 million transitions. For each RL algorithm in the default model case, we train three separate agents with different seeds. For the model ablation case, we train only one agent due to computational cost.

**World Model:** The world model utilizes the *dual network* architecture proposed in Thinker, as it enables the prediction of raw states, values, and policies, allowing its use in both simulation-based approaches and planning in MuZero and Thinker. We also found that using the dual network results in better performance in the environment than the original network proposed in MuZero, likely due to the addition of learning signals from predicting the raw state. In the small model ablation case, we reduced all channel sizes in the RNN block from the default 128 to 32.

We follow the training procedure discussed in Thinker to train the world model, except that we added an additional loss based on L2-distance between the predicted raw state and the true raw state. This ensures that the world model focuses on all features of the raw states, not just those relevant to rewards. Consequently, non-reward-affecting features, such as the blue location, can still be encoded and predicted by the world model. We found that the addition of this loss does not negatively impact the agent's performance in the environment. It should be noted that we do not assume we have knowledge about the event $g$ when training the world model or the agent. We train a separate world model following the same training procedure for each agent.

Figure 5 shows examples of model outputs for both the default setting and the model ablation setting. In the default case, the model predicts the states accurately, albeit with slight blurring. In the small model case, the agent erroneously pushes the box across the wall, which should not be allowed, and the blue location is missing, likely due to the model's limited capacity preventing it from fitting into the representation. In the stochastic case, the agent gradually fades due to the uncertainty of its position. Lastly, in the POMDP case, the agent is completely missing, attributable to the difficulty of ascertaining the agent's true position. These three model ablation cases thus showcase the different failure modes of the model.

**Learning Curve:** The learning curves of the agents in both the default setting and the model ablation setting can be found in Figure 6. We observe that in the default setting, the Thinker agent performs the best, with results closely replicating those of the original paper. The MuZero agent here outperforms the MuZero agent in the Thinker's paper due to the use of a dual network as the world model. In the small model setting, the performance of the DRC and baseline agents is similar to that in the default case, as the world model is not used in the policy. In the POMDP case, all agents perform poorly, likely because they cannot be certain of their own location, making the problem too challenging.

# B  Predictor Details

**Predictor architecture :** The raw states and predicted states (if they exist) are processed by a separate convolutional encoder with the same architecture. The encoder shares the same architecture as follows:

---

[3]We count each node traversal as one simulation, as opposed to one new node expansion

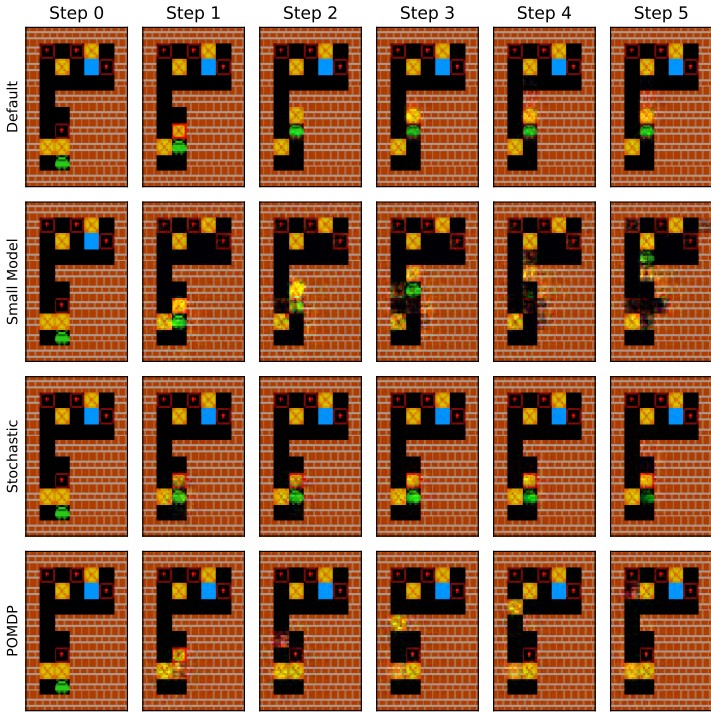

Figure 5: The predicted states output by the trained world model, where the starting state is shown in the leftmost column and the input action is five consecutive UP actions.

- Convolution with 64 output channels and stride 2, followed by a ReLu activation.
- 1 residual blocks, each with 64 output channels.
- Convolution with 128 output channels and stride 2, followed by a ReLu activation.
- 1 residual blocks, each with 128 output channels.
- Average pooling operation with stride 2.
- 1 residual blocks, each with 128 output channels.
- Average pooling operation with stride 2.

All convolutions use a kernel size of 3. The resulting output shape is (128, 6, 6). The output is then flattened and passed to a linear layer with an output size of 128. The encoded state is then concatenated with the action selected on that state to form an embedding. For the inner state approach of Thinker, we also concatenate the tree representation [6] to the embedding.

For the simulation-based approach applied to all agents and the inner state approach used by explicit planning agents, the embedding of the current state combined with the rollouts forms a sequence of embeddings of size 1 + total rollout length. For the inner state approach on the DRC agent, we encode the hidden state of each internal tick using the same convolutional encoder mentioned above, but without average pooling. Since there are four internal ticks ($t = 0, 1, 2, 3$) in DRC(3,3), they form a sequence of embeddings of size four. This sequence is concatenated with the current state embedding to form a sequence of five embeddings. Similarly, for the inner state approach on the IMPALA agent, we encode the hidden layer using the same convolutional encoder but also without average pooling. This is concatenated with the current state embedding to form a sequence of two embeddings. Note that the embedding of the current state is always positioned at the first slot in the sequence.

In all cases, the sequence of embeddings passes through a three-layer Transformer encoder with a dimension of 512. The output from the Transformer encoder at the first token is then passed to a linear layer, which predicts the required probabilities using either softmax or sigmoid output units.

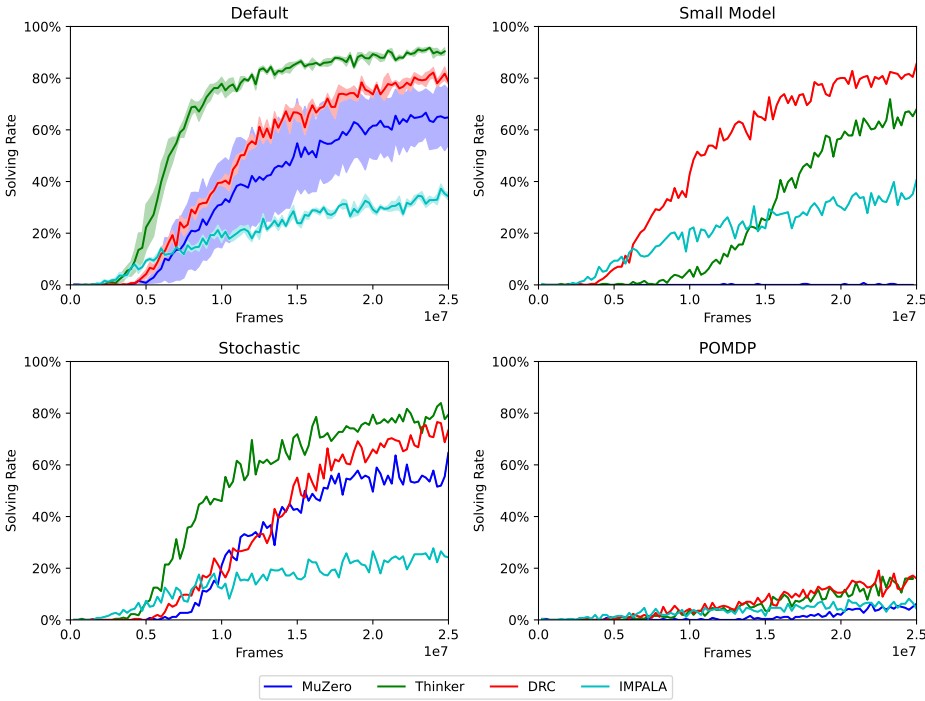

Figure 6: Running average solving rate over the last 200 episodes in Sokoban in both the default setting and the model ablation settings. For the default case, the shaded area represents two standard errors across 3 seeds.

**Training:** We generate 50,000 training samples, 10,000 evaluation samples, and 10,000 testing samples using the trained agents. We perform stochastic gradient descent on the cross-entropy loss to train both the action predictors and incident predictors. We utilize a batch size of 128 and an Adam optimizer with a learning rate of 0.0001. Training is halted when the validation loss fails to improve for 10 consecutive steps. For each agent with a unique seed, we train three independent predictors. As we have three separately trained agents for the default model case, this leads to a total of 9 runs.

# C Experiment Details

Figure 7 shows the final accuracy of action prediction and the F1 score of incident prediction for the model ablation settings. The change in performance shown in Figure 4 is computed as the difference between the performance in the default model case and the performance shown here in Figure 7.

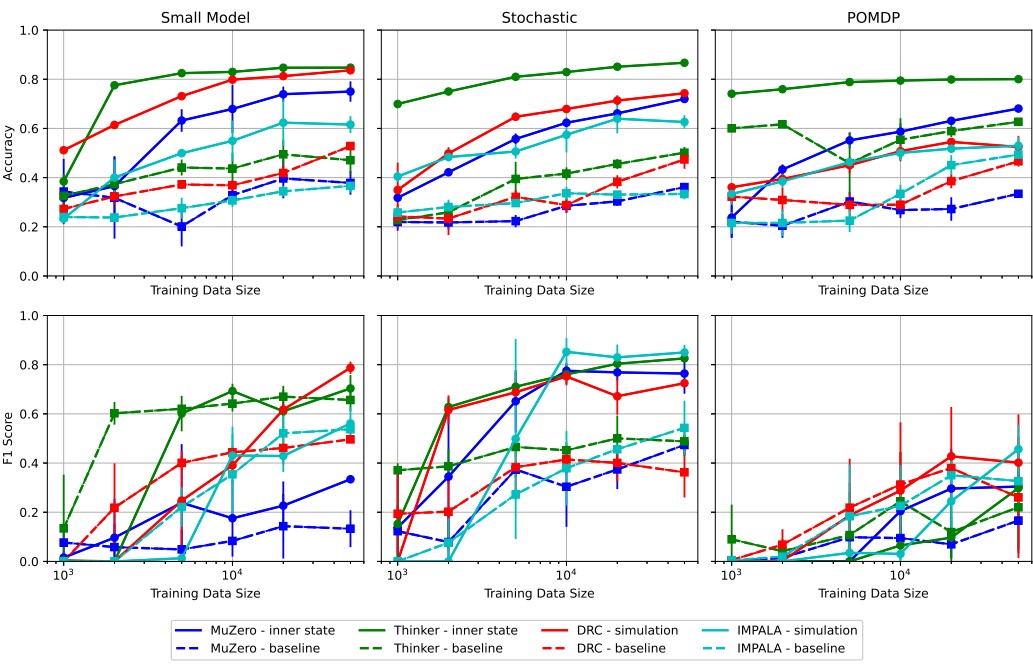

Figure 7: The final accuracy of action prediction and F1 score of incident prediction for the world model ablation settings. The error bar represents two standard errors across 3 seeds.

**Computational Resources:** Each agent is trained using a single A100 GPU, with training time varying by algorithm. MuZero, Thinker, DRC, and IMPALA take approximately 7, 3, 2, and 1 days, respectively, to complete training. The world model is trained concurrently with the agent on the same GPU. For training the predictors, we also use a single A100 GPU, and it takes about 2 days to complete training across all auxiliary information settings for a single agent and a single seed.

**Code:** The code used for these experiments is available at `https://github.com/stephen-chung-mh/predict_action` and is based on the public code released in Thinker [6].

# D    Ablation on Inner State Approach

We consider alternative choices for inner states and repeat the experiment shown in Figure 2. The following inner states are considered:

1. MuZero: We considered using the top 3 rollouts ranked by visit counts against only the top rollouts (the default case).

2. DRC: We considered using the hidden state at all ticks (the default case) against only the hidden state at the last tick.

3. IMPALA: We considered using the output of all three residual blocks against only the last residual block (the default case).

The results can be found in Fig 8. We observe that the results are similar with different chosen inner states, except that (i) using top 3 rollouts in MuZero leads to slightly lower event prediction accuracy, possibly because the top rollout is sufficient to make the prediction, and (ii) using all residual blocks in IMPALA gives slightly better performance in event prediction, likely because lower residual blocks still encode the blue location that is helpful for predicting the chosen event.

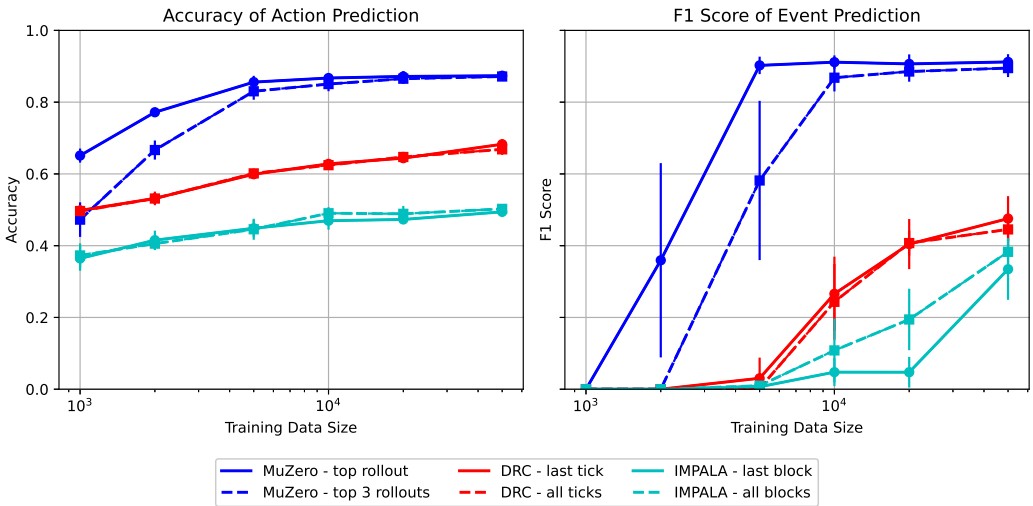

Figure 8: Ablation experiments on the chosen inner state. The error bars represent two standard errors across 9 seeds.

