# OpenReview forum: "Predicting Future Actions of Reinforcement Learning Agents"
_NeurIPS.cc/2024/Conference — NeurIPS 2024 poster_

### Official Review · Reviewer_tXgP · 2024-07-12

**Soundness:** 3
**Presentation:** 3
**Contribution:** 2
**Rating:** 5
**Confidence:** 4

**Summary:**

This paper investigates methods for predicting future actions and events of reinforcement learning agents during deployment. It compares the predictability of explicitly planning, implicitly planning, and non-planning agents using two approaches: an inner state approach that leverages the agent's internal computations, and a simulation-based approach that uses a learned world model. The result from the Sokoban environment shows that plans from explicitly planning agents are more informative for prediction than other types of inner states.

**Strengths:**

- This paper addresses an important and practical problem in RL deployment - predicting future agent behavior.
- The proposed method is well-motivated, with a clear analysis of its idea.
- The structure and presentation of the paper are clear and well-organized.

**Weaknesses:**

See the questions below.

**Questions:**

- What's the rationality behind the choices in different methods?
	- For MuZero, why only use the most visited rollout as auxiliary information? Wouldn't using top-K rollouts provide more information?
	- For DRC, have you experimented with using only the final tick or a subset of ticks?
	- For IMPALA, have you tried using multiple layers or earlier layers?
- For simulation-based approach, how do you handle the potential divergence between simulated trajectories and real trajectories over time?
- How are the training/testing samples generated? Are they from trajectories with fixed length?

---

> ### Author Rebuttal · Authors · 2024-08-06
>
> We are thankful for the reviewer’s comments and for recognizing the significance of the problem statement. The questions are addressed as follows:
>
> > What's the rationality behind the choices in different methods?
>
> The rationality is mainly based on our intuition of which information is useful. We agree that more comprehensive experiments would benefit the paper. As such, we conducted further experiments to show the impacts of choosing different types of inner-state. Please refer to our global response for the results. We will include these results in the revised paper.
>
> > How do you handle the potential divergence between simulated trajectories and real trajectories over time?
>
> First, we note that our problem statement does not permit access to real trajectories, so predictions must be made purely on the basis of simulated trajectories and the current state, which may diverge depending on the quality of the world model. We make no special effort to prevent this divergence, although we note that the world model is trained on transitions the agent encounters during training.
>
> To cope with the potential divergence when predicting future actions and events, we input the simulated trajectories to a neural network, as the neural network should learn to extract useful information from the trajectories while discarding the non-useful information. In the extreme case of the world model outputting random noise, such that the simulated trajectories contain almost no information, the neural network can still rely on the current state to make a prediction (which is equivalent to the baseline case shown in Fig. 2). We also note that it is due to this possible divergence that we do not just use the empirical distribution of simulated trajectories to manually predict future actions and events.
>
> > How are the training/testing samples generated? Are they from trajectories with fixed length?
>
> We deploy the trained agents on the training level of Sokoban until 50k transitions are collected. Each episode may last from a few steps to over a hundred steps, depending on whether and how fast the trained agent can solve the level. We perform the same process for generating testing samples, but we deploy the trained agent on testing levels instead of training levels and only collected 10k transitions.
>
> We are thankful for the reviewer’s suggestions and hope that our response adequately addresses the reviewer’s questions. We hope that the reviewer considers revising the score in light of our new experiments and clarification.

---

> > ### Comment · Reviewer_tXgP · 2024-08-12
> >
> > I appreciate the author's response, most of my concerns have been resolved. For now, I'll keep my score and continue to pay attention to other reviews and ongoing discussions.

---

### Official Review · Reviewer_W8D8 · 2024-07-12

**Soundness:** 2
**Presentation:** 3
**Contribution:** 2
**Rating:** 7
**Confidence:** 4

**Summary:**

This paper focuses on predicting future actions and events using policies learned by different types of planning agents. Namely, the authors distinguish between explicit and implicit planning agents, where explicit planning agents learn a model of the environment dynamics/transitions. Given this distinction between agents, the authors investigate how well these agents can reliably predict future actions or reaching a certain states within a finite time frame. They study the agents empirically in the domain of Sokoban and show the findings of their study.

**Strengths:**

**Originality:** This is an investigative study on how the type of agent being trained influences its ability to predict future actions and events. While the prediction of future actions is not a new problem, the novelty arises from the study over different types of agents.

**Significance:** I believe that the problem is interesting and understanding this topic can go along way in how users of AI agents can decide on which agents to train/deploy --- e.g., "does one learn an environment model or are the gains not worth the increase in complexity?", etc.

**Clarity:** The paper was clear.

**Quality:** The main motivation behind the paper is simple and well-defined. The problem being studied is an interesting one. The chosen experimental design is reasonable for what the authors are trying to show.

**Weaknesses:**

- The results are plotting standard deviation instead of standard errors to show confidence intervals.

- Predicting the event of reaching the blue tile seems like it may not be a good evaluation criteria. If the blue tile can be placed, with uniform probability, on any empty tile in the puzzle, then if the puzzle is large, the agent is very unlikely to reach the tile. This prediction task seems very trivial.

**Questions:**

See my last point in the previous section.

**Limitations:**

Yes, but these seem like things the authors should do to make the paper stronger.

---

> ### Author Rebuttal · Authors · 2024-08-06
>
> We are thankful for the reviewer’s comments and for recognizing the significance of the problem statement. The questions are addressed as follows:
>
> **W1**: As the goal of the figure is to compare the effectiveness of different approaches on different agents instead of the prediction variability, it is indeed better to use a 2-sigma error bar that indicates the confidence interval instead. We will update the paper to use 2-sigma error bar.  Please see the updated figures in our global response for the revised figures.
>
> **W2**: We designed the event to be predicted to have a small probability, so it is closer to real-world applications where we may want to predict an unsafe but rare event. In our trained agents, the agent will step on the blue tile within L=5 steps around 5% of the time. Therefore, a trivial predictor that predicts the future event not occurring will already achieve 95% accuracy. As such, we report F1 score instead of accuracy in event prediction. Such a trivial predictor will only achieve a zero F1 score due to zero recall, making the problem far from trivial. We expect that an even rarer event will make achieving a high F1 score even more challenging due to the scarcity of the data.
>
> We are thankful for the reviewer’s suggestions and hope that our response adequately addresses the reviewer’s questions. We hope that the reviewer considers revising the score in light of our clarifications.

---

> > ### Comment · Reviewer_W8D8 · 2024-08-12
> > **Response to rebuttal**
> >
> > I would like to thank the authors for their response to my concerns. As they have addressed the statistical concern and explained the other I will increase my score.

---

### Official Review · Reviewer_SSgd · 2024-07-13

**Soundness:** 2
**Presentation:** 1
**Contribution:** 2
**Rating:** 4
**Confidence:** 4

**Summary:**

The paper presents a comparative analysis of future action and event predictions in Reinforcement Learning settings by comparing inner-step approaches (that predict via inner computations like by generating a plan or via neuron activations) and simulation-based approaches (that simulate a rollout in a learned world model). The results indicate that while the latter are superior in terms of plans generated, the former methods are more robust in future action predictions.

**Strengths:**

1) The problem statement is cleanly defined, and is easy to understand the rest of the paper.
2) The paper provides a neat overview of each of the approaches compared via empirical results.

**Weaknesses:**

1) The authors mention that simulation-based approaches are superior in terms of plans generated, but inner-state approaches are more robust in future action prediction. Why is this result surprising? What was the hypothesis to begin with? Can the contribution of the paper be re-stated to empirically showing an expected result? There is no issue if that is the case, but the fact that there is no mention of an initial hypothesis to draw comparisons between the two types of approaches is misleading.
2) What is the intuition behind the categorization of planning agents into explicit, implicit and non-planning? What definition of planning do authors have in mind such that these categories are important to talk about?
3) Line 176: Why are simulation-based approaches expected to predict the same state space as the input to the agent? Works such as Dreamer and Planet, as far as I understand, learn to predict the latent states in the world model. What is the reason if this specific class of approaches are being considered?
4) Why is Section 5 Related Works in the middle of the paper? Moreover, the bigger issue I find is none of the topics of Safe RL, Opponent Modelling in MA settings, and Predictability for Human-AI interactions are relevant as "related works". Indeed, these are important applications to the approaches being compared in this work, but I am not sure how they can be counted as related works.
5) What are the final takeaways of this work? Which approaches need to be considered for which respective problem settings?

Minor comments:
1) Why is section 4 titled Methods? There are no method contributions in this work. Please rephrase it as Comparative Study or something like it.
2) Shift line 159-160 before line 145.

**Questions:**

Please see the weaknesses section for question.

**Limitations:**

1) Why do authors write that environments with diverse states do not exist? There are several RL benchmarks used for evaluating diversity in exploration. Is there any other type of 'diversity' that is being talked about here for which designing new environments will be required?
2) What is meant by 'false alarms' in line 350? Please rephrase the broader impact statement for better clarity.

---

> ### Author Rebuttal · Authors · 2024-08-06
>
> We are thankful for the reviewer’s detailed comments. The questions are addressed as follows:
>
> **W1**: Please see our global response. Briefly, our results largely accord with our hypotheses, and we will add the hypotheses to the introduction following the list of research questions.
>
> **W2**: We define planning as “the process of interacting with an environment simulator or a world model to inform the selection of subsequent actions” (lines 71-72). Our distinction is meant to capture the difference between agents that do this explicitly by construction (explicit planning agent), those that behaviorally seem to do so implicitly by learning (implicit planning agent), and those that behaviorally do not seem to do so at all (non-planning agent). The exact definition of these three types of agents can be found in lines 75-96.
>
> The type of planning is important when considering predicting future actions or events as different types of information are available. With an explicit planning agent, the rollouts in the world model provide an easily interpretable plan (e.g., we can directly look at the rollouts in Fig. 1 of the Thinker paper). With an implicit planning agent, the hidden activations may contain the same degree of information as the explicit rollouts in explicit planning agents. However, this information is distributed across neurons and likely mixed with other internal computations that are not relevant to its plans, making it harder to extract the agent's plans. Lastly, for a non-planning agent, the lack of planning means that the hidden state may only contain a more compact representation of the state. It is expected that these different agents' inner states have different degrees of information and intractability. This expectation is largely consistent with the results shown in Fig. 2.
>
> **W3**: In the simulation-based approach, we unroll a given agent in a trained world model instead of the environment. Therefore, it is expected that the trained world model predicts the same state space as the input to the agent. In special cases where the action can be computed using only the abstract state (e.g., the actor in Dreamer only requires the latent state to compute the action distribution), we can indeed use an abstract world model (e.g., the same latent world model in Dreamer) to do the simulation. We are thankful that the reviewer pointed this out and will clarify it in the revised paper.
>
> P.S. Latent predicted state may be less effective for event prediction compared to the raw predicted state. We briefly experimented with a latent world model using Thinker and found that the event prediction accuracy dropped significantly, likely because the information in latent representation is harder to extract. However, if we train a decoder to convert the latent predicted state to the raw predicted state, the accuracy restores to the level reported in the paper. As such, we advise decoding the latent predicted state into the raw predicted state even when rollouts in latent space are possible.
>
> **W4**: We will move Section 5 after the introduction to improve clarity. As aforementioned, the paper considers a new problem setting, so we are unable to find any related work that tackles the same problem. As such, we can only list the works done in related areas and/or share similar goals, such as predicting opponent actions in a multi-agent setting or preventing (instead of only detecting) dangerous agent actions in a safe RL setting.
>
> **W5**: Please see our answer to the three research questions in our global response. In particular, we advise using the simulation-based approach when the environment dynamics are easy to model, and using the inner-state approach in other cases. We also advise using explicit planning agents if the inner-state approach is considered, as their explicit plans are generally more informative or easily extractable than the inner states of other agents.
>
> > Why is section 4 titled Methods? There are no method contributions in this work.
>
> We propose two methods, and both methods are part of the paper’s contributions. As this is a new problem setting, the two methods are constructed to be general and simple to allow future work to build upon them.
>
> We note that the simulation-based approach has been used in a multi-agent setting to predict opponent movements for training a better policy, but the nature of the problem is very different from the one we consider. For the inner-state approach, we are not aware of any similar methods used in other problem settings.
>
> > Shift line 159-160 before line 145.
>
> Thank you. We will update the paper accordingly.
>
> > Why do authors write that environments with diverse states do not exist?
>
> The diversity here refers to the variety of possible solutions (or optimal policies) as in Sokoban, where there are usually multiple ways of solving the level, leading to high difficulty in predicting agent actions. We did not posit that such environments do not exist, and welcome pointers to alternative environments with this property, as we have not been able to identify any. We briefly experimented with Atari environments, where in some games the state space is very diverse, but we found that the lack of diverse optimal policies reduces the problem of future action prediction to predicting the optimal actions.
>
> > What is meant by 'false alarms' in line 350?
>
> It means that when the predictor forecasts that an agent is going to perform an unsafe action, but in fact, the agent won’t be doing it. This may lead to improper responses (such as shutting down the agent) that are not warranted. We will rephrase the broader statement to make it clearer.
>
> We are thankful for the reviewer’s detailed feedback. We will update the paper as stated above and hope that the reviewer considers revising the score in light of the clarified hypothesis and context. We are also happy to discuss further if the reviewer has any additional questions.

---

> > ### Comment · Reviewer_SSgd · 2024-08-11
> > **Response to Author Rebuttal**
> >
> > Thanks for your response! I have updated my assessment after going through the responses and the rest of the author-reviewer discussions.

---

> > > ### Author Response · Authors · 2024-08-12
> > >
> > > Thank you for the updated assessment. Could you please share the specific reasons for rejecting the paper? It appears to us that the questions raised in the weaknesses section have been addressed adequately. We would appreciate any additional feedback on how we can improve the paper.

---

> > > > ### Comment · Reviewer_SSgd · 2024-08-12
> > > > **Response to Author Rebuttal**
> > > >
> > > > #### **[W1]**
> > > >
> > > > I am still not convinced that the hypotheses and the empirical results go beyond anything currently not known about the different types of planning agents, particularly the ones considered in the paper. Errors in policy learning in 'explicit' planning agents based on an incorrectly learned world model is already well understood. Similarly, the results for 'implicit' planning agents are not surprising either. A more objective analysis, even for a single category of approaches, could make the paper much stronger.
> > > >
> > > >
> > > > #### **[W2]**
> > > >
> > > > I do not agree with the authors' definition of planning. Planning in its crude form can be understood as search, and what distinguishes planning is the fact that there are dependencies between states. The proposed taxonomy is simply making the distinction if policy learning is either based off on first learning an explicit world model, or learning latent states, or none of the above. The proposed definitions in the paper (lines 75-96) primarily are interpretations of different methods in literature. The paper does not provide any mathematical or objective definitions to distinguish between the different methods. It is not clear why this taxonomy was needed at the first place - the empirical questions of action prediction are event prediction are not sufficient to motivate this. Even if the current taxonomy is taken into consideration, it would have been much more useful to understand the reasons behind advantages/disadvantages of learning latent representations and/or a world model.
> > > >
> > > >
> > > > #### **Section 4**
> > > >
> > > > It is hard to interpret the proposed taxonomy as 'Methods' in the current version of the work. A proposed method is a formal framework and/or approach, which is not a contribution in the current version of this work. The paper provides a taxonomy, which I believe is still not objectively defined, over existing approaches and performs a comparative evaluation among them.
> > > >
> > > > In summary, the paper's claims, writing, presentation, and structure, all require significant changes to make it a stronger contribution.

---

> ### Author Response · Authors · 2024-08-12
>
> Thank you for the response and the feedback.
>
> > I am still not convinced that the hypotheses and the empirical results go beyond anything currently not known about the different types of planning agents, particularly the ones considered in the paper. Errors in policy learning in 'explicit' planning agents based on an incorrectly learned world model is already well understood. Similarly, the results for 'implicit' planning agents are not surprising either. A more objective analysis, even for a single category of approaches, could make the paper much stronger.
>
> We respectfully disagree with the assertion that the hypothesis and empirical results are already known. To the best of our knowledge, there has not been any study on the predictability of different RL agents. We would appreciate any references to relevant papers if they exist. Our experimental results partially confirm our hypothesis, while some findings are relatively more surprising (as stated in the global response). It’s important to note that the error in policy learning is distinct from what we are considering here—we are investigating the predictability of RL agents, not the process of learning a policy.
>
> > I do not agree with the authors' definition of planning.
>
> We did not propose any new definitions for planning or a new taxonomy of RL agents in the paper. Our definition of planning is consistent with previous work, such as [1] and [2]. There are many proposed definitions of planning. We do not believe there is a universally agreed-upon definition.
>
> Recent works [3, 4] suggest that agents can also learn to perform planning with the correct architecture. Therefore, we believe that further distinguishing between explicit and implicit planning agents is beneficial for discussing the results, as labeling them as planning-free contradicts the findings of these works. This distinguishment between explicit and implicit planning agents is not novel, as it has already been addressed in the literature—see, for example, Table 5 in [5], which distinguishes between explicit and implicit planning. Developing a rigorous mathematical definition for explicit and implicit planning agents is far from trivial (and was also not attempted in [5]), as it is difficult to determine whether an agent learns to plan within its neural network. This challenge is well beyond the scope of our paper.
>
> In summary, our classification of agents follows existing work, and we have not introduced any new definitions or made any claims to contribute to the taxonomy of existing RL agents. We will reference [5] in the revised paper when discussing the agent types to clarify that this is not a new taxonomy. We included the taxonomy solely to aid in discussing the methods and results.
>
> [1] Chung, S., Anokhin, I., & Krueger, D. (2024). Thinker: learning to plan and act. Advances in Neural Information Processing Systems, 36.
>
> [2] Hamrick, J. B., Friesen, A. L., Behbahani, F., Guez, A., Viola, F., Witherspoon, S., ... & Weber, T. On the role of planning in model-based deep reinforcement learning. In International Conference on Learning Representations 2021.
>
> [3] Guez, A., Mirza, M., Gregor, K., Kabra, R., Racanière, S., Weber, T., ... & Lillicrap, T. (2019, May). An investigation of model-free planning. In International Conference on Machine Learning (pp. 2464-2473). PMLR.
>
> [4] Garriga-Alonso, A., Taufeeque, M., & Gleave, A. (2024). Planning behavior in a recurrent neural network that plays Sokoban. In ICML 2024 Workshop on Mechanistic Interpretability.
>
> [5] Moerland, T. M., Broekens, J., Plaat, A., & Jonker, C. M. (2023). Model-based reinforcement learning: A survey. Foundations and Trends® in Machine Learning, 16(1), 1-118.
>
> > It is hard to interpret the proposed taxonomy as 'Methods' in the current version of the work. A proposed method is a formal framework and/or approach, which is not a contribution in the current version of this work.
>
> Section 4 proposes using inner-state and simulated rollouts to aid in predicting future actions and events. We introduced new types of additional information that can enhance prediction and experimentally demonstrated that they indeed improve prediction accuracy. A taxonomy typically involves a simple categorization of existing methods, but we have emphasized that both of these methods are novel. Simultaneously proposing two methods in a new problem setting does not constitute creating a taxonomy. We strongly disagree with the reviewer’s assessment that our two proposed methods should be dismissed as merely a taxonomy.

---

> > ### Comment · Reviewer_SSgd · 2024-08-12
> > **Response to Author Rebuttal**
> >
> > Thanks once again for your responses and respectful disagreements! I very well understand the authors’ perspective, and this discussion has surely been helpful to me to get a better understanding of the idea behind the work. I’ll maintain my current assessment and my incremented score.

---

### Official Review · Reviewer_7yyB · 2024-07-14

**Soundness:** 2
**Presentation:** 3
**Contribution:** 2
**Rating:** 6
**Confidence:** 3

**Summary:**

The authors present two approaches for predicting future states and actions in an RL setting. This is done in a supervised setting, where the inputs are the state and action and the target is either the future action sequence or the time of occurrence of some event. The first approach leverages using the inner-state as additional information to the state action input. The second approach uses a model trained to simulate the environment to generate a rollout to serve as the additional information.

**Strengths:**

While look-up tables of reward-action have been utilized before, the reviewer is not aware of any prior work that includes entire rollouts, or leverages external hidden states as predictors of future events or actions. The authors’ method is clearly explained. The experiments clearly demonstrate the authors’ claims and both metrics illustrate the capability of their approach in predicting future actions and events. The problem statement, prediction of future actions and the events that may be a result, is undoubtably a significant issue in the larger ML community as a whole.

**Weaknesses:**

The reviewer’s primary concern is with data leakage that could occur with the rollouts used as additional information when training the model, especially with the ‘Thinker’ agent. If the action space is rolled out and appended to the input, would this not result in the potential future actions already being included with the input?

The reviewer is also unsure as to the benefits of the method over, for example, training a world model, rolling out several series of actions and choosing the branch that accumulates the highest predicted reward to also serve as a predictor of future actions.

**Questions:**

The reviewer invites the authors to kindly clarify if they have misunderstood any parts of the paper leading to the weaknesses outlined above. Beyond those related to the weaknesses, the reviewer does not have any other questions.

**Limitations:**

The authors acknowledge the limited range of RL agents and environments tested.

---

> ### Author Rebuttal · Authors · 2024-08-06
>
> We are thankful for the reviewer’s comments and for recognizing the significance of the problem statement. The questions are addressed as follows:
>
> > If the action space is rolled out and appended to the input, would this not result in the potential future actions already being included with the input?
>
> These rollouts are rollouts of imaginary actions performed in a learned world model and thus do not constitute data leakage. This allows action and event predictions to be made L=5 steps before the true future action or event is known. The future action may indeed be included in simulated rollouts of model-based agents, but these agents (including Thinker) won't **necessarily** choose the actions that are simulated in the rollouts due to numerous reasons, such as an agent learning to ignore bad rollouts, adjusting for model prediction error, or continuously refining the plan in future steps. Nonetheless, the simulated actions in rollouts are certainly helpful for predicting future actions—this is expected, and this potential advantage of explicit planning algorithms is one of the core motivations of our work.
>
> > The reviewer is also unsure as to the benefits of the method over, for example, training a world model, rolling out several series of actions and choosing the branch that accumulates the highest predicted reward to also serve as a predictor of future actions.
>
>
> We interpret the suggestion as using the empirical action or event distribution of the rollout with the highest rollout returns (sum of predicted rewards and the terminal value) as the predicted future action and event distribution.
>
> For model-based RL agents such as Thinker and MuZero, they do not necessarily select the rollout with the highest rollout return. Figure 19 in the appendix of the Thinker paper shows that a trained Thinker agent selects the next action with the highest rollout return around 90% of the time. In contrast, the same figure for MuZero (shown as MCTS) is around 50%, as MCTS selects the next action based on visit count instead*.
>
> For model-free RL agents, as they cannot perform rollouts like Thinker or MuZero, their selected actions likely differ from the best action from multiple rollouts. They often exhibit myopic behavior, such as pushing a box to a wall and thus making the level unsolvable. The empirical rollouts (i.e., rollouts using the agent policy) should be more insightful for predicting future actions and events.
>
> As such, if we had to estimate the future action or event prediction by empirical distribution of a rollout, we suggest using**:
>
> 1. The rollout with the highest return for Thinker.
> 2. The rollout with the largest visit count for MuZero.
> 3. The empirical rollout for model-free RL agents.
>
> However, in our paper, we consider inputting these rollouts to a neural network to learn the prediction instead of directly using the empirical distribution. This is because the world model may diverge from the true environment, leading to unreliable rollouts. For example, in the extreme case that the trained world model outputs random noise, the learned neural network can still predict future actions or events based on the current state while ignoring the rollout, resulting in the baseline case shown in Figure 2. This should be considerably better than using the empirical distribution of rollouts, which does not provide any information in this hypothetical example.
>
> Another advantage of using a learned neural network is that we only need the binary event label instead of the event function when training the predictor. On the other hand, if we use the empirical distribution of rollouts to make event predictions, we also need to handcraft a function that maps predicted states to an event label (e.g., classify whether the agent stands in the blue location from raw predicted pixels in our setting).
>
> *Note that both figures will be lower if we consider the action in L steps (which we are interested in predicting) instead of just the next action.
>
> **The first two are included in the inner-state approach, while the third is included in the simulation-based approach.
>
> We hope that the above answer clarifies the reviewer’s questions and look forward to further discussion.

---

### Author Rebuttal · Authors · 2024-08-06

We are thankful for the reviewer’s comments. Before addressing the questions, we would like to clarify the context of the paper. In this paper, we consider a problem statement, namely predicting future actions and events for agents trained with different RL algorithms. As far as we are aware, this problem has not been considered in the previous literature, despite most reviewers recognizing its significance. We propose two methods for this problem, namely the inner-state approach and the simulation-based approach. Despite their simplicity, both methods have not been studied before in this context. We then evaluate and compare the two approaches empirically to answer the following three research questions listed in lines 40-48:

- **Q1.** How informative are these inner states for predicting future actions and events?
- **Q2.** How does the predictability of future actions and events vary across different types of RL agents with different inner states?
- **Q3.** How do the performance and robustness of the simulation-based approach compare to the inner state approach in predicting future actions and events across different agent types?

Before performing the experiments, we had some hypotheses about the answers to the above questions. We also summarized the results here to aid understanding.

**Q1 and Q2**

**Hypothesis**: The benefits of the inner state depend on the type of agent. The inner state for the explicit planning agent should be more informative than that of the implicit planning agent, as the plan in the explicit planning agent is more interpretable (the rollouts are human-interpretable) while the plan for the implicit planning agent is stored in the hidden activation. Nonetheless, they both contain indications of the agent’s future actions, so they should hold more information than the inner state of a non-planning agent, which does not contain future plans and may only be a more compact representation of the state.

**Result**: For future action prediction, these hypotheses are consistent with the results shown in Fig. 2, where the inner-state approach performance of the explicit planning agent is better than that of the implicit planning agent, followed by non-planning agents (line 257-270). For future event prediction, the result is more surprising as the inner-state for both implicit planning agents and non-planning agents does not help in event prediction. We conjecture that the agent ignored our defined events in its representation as they do not affect rewards. On the other hand, the inner state of explicit planning agents is still very beneficial, as the world model is explicitly trained to predict all features in the environment (lines 271-277).

**Q3**

**Hypothesis**: If we have an accurate world model, simulation-based approaches should be better. Consider the extreme case of having a perfect world model; then the exact future action and event distribution can be estimated by the empirical distribution of the rollouts. However, we expect the simulation-based approach to be much less robust to model quality than the inner-state approach, as a slight error in the predicted state may confuse the agent and make it act differently due to out-of-distribution (OOD) inputs. For the inner-state approach, in the case of a model-free RL agent, there is no model, so this is not directly comparable. And in the case of a model-based RL agent, the agent is trained along with the model, so even inaccurate model output is not OOD to the agent, possibly resulting in better robustness.

**Result**: Fig. 3 shows that the simulation-based approach indeed performs better than the inner-state approach, as our trained world model is close to perfect. However, when we do ablation on the world model, as shown in Fig. 4, the performance of the simulation-based approach generally drops more for action prediction, which is consistent with our hypothesis (line 314-322). For event prediction, the result is more mixed and surprising, with the simulation-based approach sometimes performing better and sometimes worse. We conjecture that this is due to the trained world model not predicting the defined events well, leading to poor prediction in both approaches (lines 323-327).

We did not put our initial hypotheses in the introduction, as similar discussions are already included in the discussion section. However, we agree with reviewer SSgd that stating the initial hypotheses at the beginning will improve the clarity and flow of the paper, and we will add that to the revised version of the paper.

To improve the robustness of the results, we have repeated the main experiments with 9 seeds and will update the main figures of the paper by plotting the 2-sigma error bar instead of the standard deviation. The updated figures can be found in the attached PDF. In addition, we performed an additional ablation study on the chosen inner state:

- **MuZero**: We considered using the top 3 rollouts ranked by visit counts against only the top rollouts (the default case).
- **DRC**: We considered using the hidden state at all ticks (the default case) against only the hidden state at the last tick.
- **IMPALA**: We considered using the output of all three residual blocks against only the last residual block (the default case).

The results can be found in Fig A3 of the attached PDF. We observe that the results are similar with different chosen inner states, except that (i) using top 3 rollouts in MuZero leads to slightly lower event prediction accuracy, possibly because the top rollout is sufficient to make the prediction, and (ii) using all residual blocks in IMPALA gives slightly better performance in event prediction, likely because lower residual blocks still encode the blue location that is helpful for predicting the chosen event.

We are thankful for the reviewers’ detailed feedbacks and suggestions.

---

### Decision · Program_Chairs · 2024-09-25

**Decision:**

Accept (poster)

**Comment:**

Nearly all reviewers agreed that this work is original, significant, clear, and well-motivated. Any lingering questions or issues were resolved during the rebuttal. In light of this, I am recommending the paper be accepted.